# A contraction approach to dynamic optimization problems

**Leif K. Sandal** [ID]**[1]**, **Sturla F. Kvamsdal** [ID]**[2]\***, **José M. Maroto[3,4]**, **Manuel Morán[4,5]**

**1** Dept. of Business and Management Science, NHH Norwegian School of Economics, Bergen, Norway, **2** SNF–Centre for Applied Research at NHH, Bergen, Norway, **3** Departamento de Economía Financiera y Actuarial y Estadística, Universidad Complutense, Madrid, Spain, **4** Departamento de Análisis Económico y Economía Cuantitativa, Universidad Complutense, Madrid, Spain, **5** IMI–Institute of Interdisciplinary Mathematics, Universidad Complutense, Madrid, Spain

\* sturla.kvamsdal@snf.no

**Data Availability Statement:** All relevant data are within the paper and its Supporting Information files.

**Funding:** Kvamsdal received financial support from the Research Council of Norway (grant no. 257630

## Abstract

An infinite-horizon, multidimensional optimization problem with arbitrary yet finite periodicity in discrete time is considered. The problem can be posed as a set of coupled equations. It is shown that the problem is a special case of a more general class of contraction problems that have unique solutions. Solutions are obtained by considering a vector-valued value function and by using an iterative process. Special cases of the general class of contraction problems include the classical Bellman problem and its stochastic formulations. Thus, our approach can be viewed as an extension of the Bellman problem to the special case of non-autonomy that periodicity represents, and our approach thereby facilitates consistent and rigorous treatment of, for example, seasonality in discrete, dynamic optimization, and furthermore, certain types of dynamic games. The contraction approach is illustrated in simple examples. In the main example, which is an infinite-horizon resource management problem with a periodic price, it is found that the optimal exploitation level differs between high and low price time intervals and that the solution time paths approach a limit cycle.

## 1. Introduction

Periodicity is an important characteristic of many systems that are subject to control. A rigorous treatment of periodicity in optimization problems is nontrivial because periodicity is a special case of nonautonomy [1]. Nonautonomy typically renders many optimal control problems difficult and costly to deal with, or even intractable. Thus, periodicity in applied work is often abstracted from altogether, or treated by considering the aggregate or mean forcing. For example, in many natural resource management models where natural growth is described as an annual process—modeled by a growth operator that is applied once per year—and where environmental conditions are known to have significant seasonal variations, an implicit assumption is that seasonal effects are aggregated up or averaged out [2]. But as shown in various research, modeling seasonal or periodic effects properly can lead to surprising and operationally important results [1, 3, 4].

To the best of our knowledge, periodicity in infinite-horizon optimal control problems in discrete time has not been treated formally in the literature. A large class of periodic problems

and no. 302197). Maroto recieved financial support from Santander Bank - Universidad Complutense de Madrid (grant no. PR108/20-14).

**Competing interests:** The authors have declared that no competing interests exist.

is a special case of a general class that can be shown to be fix-point problems for a family of contraction operators. The contraction operator can be used to obtain the solution in an iterative procedure. The class of problems considered in this study includes the classical Bellman problem, the periodic problem formulations of initial interest [1], stochastic problems, and other, more esoteric formulations. Our key contribution is nevertheless an extension of the classical Bellman result in the special case of nonautonomy that periodicity represents.

Let us first clarify the term periodicity. By periodicity, or periodic characteristics or features of optimization problems, we presently mean conditions or structures that repeat themselves at given time intervals. These recurring conditions or structures are represented by an objective function or in the description of how the state develops over time. Perhaps the simplest illustration of such periodicity is the seasonal variation inherent in many natural systems, where growth varies over a year but where the same growth conditions arise repeatedly. But note that periodic features are not necessarily described by some trigonometric relationship, and neither are solutions periodic in any other sense than that the decision rule is the same when the same conditions arise. That is, the structure of the problem repeats itself at given time intervals. The classical discrete, infinite-time dynamic programming problem in economics has discounting as the only nonautonomic feature; that is, time enters explicitly only through the discounting of the objective function. This problem represents a problem with period 1 in our setting, which is reflected by the fact that the running value function does not explicitly depend on time.

To motivate our study of periodicity in decision problems, let us briefly mention some applied examples. These examples include demand systems subject to supply control. In particular, annual, seasonal, weekly, or daily cycles in demand are well-known for electricity [5] and energy in general, and a broad range of consumer goods have seasonal fluctuations in demand. For example, McClain and Thomas [6] considered production planning under seasonal demand, whereas Bradley and Arntzen [7] further considered inventory policy and capacity constraints. More recently, Nagaraja *et al.* [8] provided a brief review of the theory related to seasonal demand problems and applied it to the bullwhip effect in supply chains. There is also an extensive literature on cyclical pricing policies discussed by Besbes and Lobel [9]. Other dynamic decision problems with periodic features are found in transport and logistics systems subject to routing control (Liebchen [10] discussed the use of optimization in the periodic event-scheduling problem) or natural systems subject to management control. For example, renewable natural resources may exhibit periodicity in growth or other natural processes, as well as in prices and costs [11, 12]. In particular, Ni and Sandal [4] studied a multiseason, multistate bioeconomic model. Kvamsdal *et al.* [1] provided a generic treatment of periodicity in resource management problems, which can be considered a special case of our results below.

To illustrate our approach to periodic problems, we apply our derived numerical scheme to a stylized decision problem with periodicity in the objective function. This example demonstrates the feasibility of our approach and suggests significant, practical implications of explicitly accounting for periodicity. In particular, the solution of the periodic problem exhibits features that are not typically present in problems with no periodicity. We also present a solution to a simple dynamic game to further illustrate the generality of our approach.

Given the prevalence of periodic characteristics of many systems subject to control, we believe that our contribution is important and highly valuable. We show that the classical Bellman problem approach can be extended to periodic problems and that this extension is, while nontrivial, both conceptually and numerically feasible and practical. Ultimately, a broader class of problems can be treated using our approach. However, both the Bellman and periodic problems are directly applicable to real-world decision problems, and thus, we maintain our focus on these formulations. Furthermore, as the periodic problem is the motivation behind

considering the problems we target in our most general result, we begin our analysis by showing how the general problem formulation suggests itself from the periodic problem setup.

The remainder of our paper is organized as follows. In the next section, we first set up a standard infinite-horizon, discrete-time optimization problem. We then generalize the problem formulation to allow for periodic variation in the problem structure and establish a set of equations that govern the problem solution. The set of equations is a special case of a general class of problems that we show are contraction problems whose solutions can be obtained through iterations. In the following section, we offer two illustrative examples: A periodic optimization problem and a dynamic game. In the dynamic game example, the notation is interpreted differently, but the equations governing the problem solution belong to the general class that we consider. In the final section, we summarize and discuss potential applications of our generalized problem formulation.

## 2. A contraction operator for the periodic problem

Dynamic decision problems under various periodic variations are our primary problem type of interest in this paper, and we begin our analysis by showing how the general problem formulation suggests itself from the periodic problem setup. We arrive at the periodic problem setup by allowing for periodic variations in a standard, infinite-horizon, discrete-time optimization problem.

A deterministic, infinite-horizon, autonomous, discounted, discrete-time optimization problem considers the following:

$$\max_{\{u_k\}_{k=0}^{\infty}} \sum_{k=0}^{\infty} \beta^{k+1} \cdot \Pi(x_k, u_k) \tag{1}$$

such that $x_{k+1} = F(x_k, u_k)$, $u_k \in U(x_k)$, $k = 0, 1, 2, \ldots$, and $x_0 \in X$ given. For the discount factor $\beta$, we have $0 < \beta < 1$. $X \subset \mathbb{R}^n$ is a feasible state space, and $x_k \in X$ is an $n$-dimensional dynamic state variable at the beginning of time interval $k$. Here, we use the term "interval" rather than "period," and reserve the latter to denominate the periodic length characteristic (denoted $T$, see below). $U : X \to \mathbb{R}^p$ is a nonempty and compact valued correspondence that specifies the admissible $p$-dimensional controls $u_k$ in state $x_k$. That is, $u_k$ is the decision or control variable that must be decided for each instant of the infinite time sequence $\{t_0, t_1, t_2 \ldots\}$. $\Pi : X \times U \to \mathbb{R}$ is bounded and continuous and gives the performance measure (return) at the end of each interval. $F : X \times U \to X$ is a continuous operator that governs the state variable such that $x_{k+1} = y_k$ is the state at the beginning of interval $k = 1$. Under these conditions, optimal controls $\{u_k^*\}_{k=0}^{\infty}$ and corresponding paths $\{x_k^*\}_{k=0}^{\infty}$ exist, as does the value function $V(x) = \sum_{k=0}^{\infty} \beta^{k+1} \cdot \Pi(x_k^*, u_k^*)$ with $x_0^* = x$. The value function is the unique fix-point of the Bellman operator $\mathcal{T}_B$, which is defined on the space of real, bounded, and continuous functions on $X$, denoted $BC(X)$, and given by

$$\mathcal{T}_B V(x) = \max_{u \in U(x)} \{\beta \cdot \Pi(x, u) + \beta \cdot V(y)\} \tag{2}$$

with $V \in BC(X)$ and $y = F(x, u)$. Using the operator defined in Eq (2), the Bellman equation for the problem in Eq (1) can be written simply as

$$V(x) = \mathcal{T}_B V(x) \tag{3}$$

See Bertsekas [13] for a more general treatment of the type of problems presented in Eq (1).

We now consider a nonautonomous but periodic problem where $\Pi_k(x, u)$ is the return function and $F_k(x, u)$ is the time evolution operator for interval $k$. That is, the return function

and the time evolution operator may vary between intervals. Sets for the feasible states ($X_k \subseteq X$) and admissible controls ($U_k$) may also vary between intervals. The control set may vary with the state such that $U_k = U_k(x_k)$, but we typically omit the state argument. The problem is periodic in the sense that for a finite integer $N \geq 1$ and for all $k \in \mathbb{N}$, we have $\Pi_k = \Pi_{k+N}$, $F_k = F_{k+N}$, $X_k = X_{k+N}$, and $U_k = U_{k+N}$. We say that the problem is periodic with period $N$, where $N$ is the smallest integer satisfying these equalities, and that the performance or return measure and the dynamic constraint functionally repeat themselves. Each period comprises $k$ intervals. The classical outset yielding Eq (1) is then a problem with period 1.

Without adding significant complexity, we can allow for varying interval lengths. Thus, each interval has potentially different discount factor values. We write the length of interval $k$ as $T_k = t_k - t_{k-1}$ and its discount factor as $\beta_k$. Periodicity implies $T_k = T_{k+N}$ and $\beta_k = \beta_{k+N}$. Then, the length of the cycle of $N$ intervals can be expressed as

$$T = \sum_{i=1}^{N} T_i = t_N - t_0 \tag{4}$$

The discount factor for the cycle of $N$ intervals is $\beta = \prod_{i=1}^{N} \beta_i$. Here, $\prod_{i=1}^{N}(\cdot)$ is the usual product operator, unrelated to the objective function elsewhere denoted by $\Pi_k(\cdot)$. Fig 1 accounts for interval index references.

Although a real discounted problem cannot have a periodic present value, the running value will be periodic under time discounting if $\beta_k$, involved operators ($\Pi_k$, $F_k$), or spaces ($U_k$, $X_k$) are periodic, as described above. As suggested above, a periodic feature repeats itself with some inherent period. If a problem includes several periodic features, the problem period $N$ will be the least common multiple of the potentially different inherent periods of the different features.

The periodic problem intuitively suggests a set of $N$ nested equations (see S1 Appendix):

$$\begin{aligned}
V_k(x) &= \max_{u_k \in U_k(x)} \{\beta_k \Pi_k(x, u_k) + \beta_k V_{k+1}(x')\}, \quad k = 1, \ldots, N-1 \\
V_N(x) &= \max_{u_N \in U_N(x)} \{\beta_N \Pi_N(x, u_N) + \beta_N V_1(x')\}
\end{aligned} \tag{5}$$

In equation set (5), $x' = F_k(x, u_k)$ is a shorthand notation for the state variable one interval ahead. If $V_k$ is interpreted as the value function for interval $k$, the equation set (5) follows from value additivity with its inherent economic logic that the present value is what you earn presently plus the discounted value of future earnings. "Earn" is not necessarily meant in its strict, monetary sense, but can be any type of utility-like flow.

In what follows, we first define generalized operators for deterministic and two stochastic formulations of optimization problems, of which the periodic problems discussed above are special cases. We then present a theorem that holds for all generalized formulations. Finally, we present a corollary that applies our theorem to the periodic problem in equation set (5).

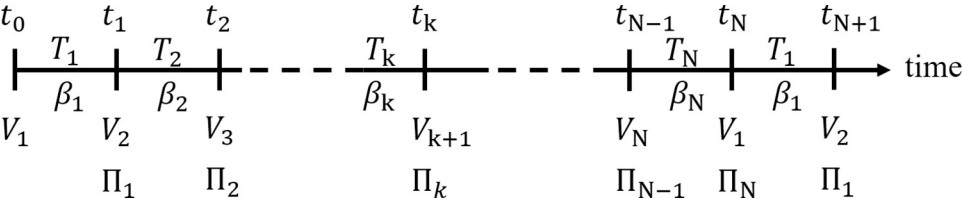

**Fig 1. Interval index reference for periodic problems.** Return ($\Pi_k$) is yielded at the end of interval $k$, whereas the value $V_k$ refers to the beginning of interval $k$.

First, we consider functional equations of the type

$$\mathcal{V}(x) = \mathcal{T}\mathcal{V}(x) \tag{6}$$

where $\mathcal{V}(x)$ is an $N$-dimensional bounded vector function in $BC(X)$ (that is, the components are bounded and continuous), and further, $x \in X \subset \mathbb{R}^n$. The components of the operator $\mathcal{T}$ are defined as

$$\mathcal{T}_k\mathcal{V}(x) \triangleq \max_{u \in \Gamma_k(x)} \{\widehat{\Pi}_k(x, u) + \beta_k \mathcal{L}_k \mathcal{V}(x')\}, \quad k = 1, \ldots, N \tag{7}$$

In (7), $\mathcal{L}_k$ are Lipschitz continuous with Lipschitz constants $\gamma_k$, $\widehat{\Pi}_k(x, u)$ are bounded functions, and the correspondence $\Gamma_k(x)$ specifies admissible sets. $\widehat{\Pi}_k(\cdot)$ can take on two forms depending on the timing of the return. If returns are realized at the end of each interval, as in equation set (5), they are discounted and we have $\widehat{\Pi}_k(x, u) = \beta_k \Pi_k(x, u)$. In contrast, if returns are realized at the beginning of each interval, they are not discounted and we have $\widehat{\Pi}_k(x, u) = \Pi_k(x, u)$. We have $\beta_k \in (0, 1)$. As we will argue below, equation set (5) is a special case of (6) when $\mathcal{T}$ is defined by (7).

Furthermore, the definition in Eq (7) is a special case of two different stochastic formulations. Let $z \in Z \subseteq \mathbb{R}^q$ be a real-valued, q-dimensional vector of stochastic elements that are realizations of a known, stochastic process (that is, the probability transition function is known and the expectation operator over $z$, denoted $E_z$, is well-defined; see Stokey *et al.* [14], p. 241). The stochastic elements can be present in both the return functions and the operators governing the state variables that are considered to be Markov decision processes. We thus write $\Pi_k(x, u, z)$ and $F_k(x, u, z)$, both of which are measurable. If both present and future realizations of the stochastic process are uncertain, we consider the following definition of $\mathcal{T}$:

$$\mathcal{T}_k\mathcal{V}(x) \triangleq \max_{u \in \Gamma_k(x)} E_z\{\widehat{\Pi}_k(x, u, z) + \beta_k \mathcal{L}_k \mathcal{V}(x')\}, \quad k \in \{1, \ldots, N\} \tag{8}$$

The definition in Eq (8) aligns with the typical formulation in Bertsekas [13].

Other problem formulations, however, consider the present realization of the stochastic process as known. Such formulations require the stochastic elements to be considered as part of the state vector. We consider $s = (x, z)$ as an extension of the state variable and consider the following definition of $\mathcal{T}$:

$$\mathcal{T}_k\mathcal{V}(s) \triangleq \max_{u \in \Gamma_k(s)} \{\widehat{\Pi}_k(s, u) + \beta_k E_{z'} \mathcal{L}_k \mathcal{V}(s')\}, \quad k \in \{1, \ldots, N\} \tag{9}$$

In (9), $s'$ is a shorthand for the extended state one interval ahead, and $E_{z'}$ is the expectancy over possible realizations of the stochastic elements in the next interval, $z'$. The definition in (9) aligns with the typical formulation in Stokey *et al.* [14]. By inspection, we see that (7), the deterministic case, is a special case of both (8) and (9). The following theorem holds for all these potential definitions of $\mathcal{T}$, that is, (7)–(9).

**Theorem:** $\mathcal{T}$ is a contraction operator on bounded vector functions if

$$\eta \triangleq \max\{\beta_k \gamma_k | k = 1, \ldots, N\} < 1 \tag{10}$$

*Proof*: Let $\mathcal{V}(x)$ and $\mathcal{W}(x)$ be arbitrary elements in $B(X)$, which is the space of bounded vector functions over $X$, and let $\|\cdot\|$ denote the sup-norm. If we write $\mathcal{L}'_k = \beta_k E_{z'} \mathcal{L}_k$, we have, for

component $k$,

$$
\begin{aligned}
\mathcal{T}_k \mathcal{V} &= \mathcal{T}_k(\mathcal{W} + \mathcal{V} - \mathcal{W}) \\
&= \max_{u \in \Gamma_k(s)} \{ \widehat{\Pi}_k(s, u) + \mathcal{L}_k' \mathcal{W}(s') + \mathcal{L}_k' \mathcal{V}(s') - \mathcal{L}_k' \mathcal{W}(s') \} \\
&\leq \mathcal{T}_k \mathcal{W} + ||\mathcal{L}_k' \mathcal{V} - \mathcal{L}_k' \mathcal{W}|| \\
&\leq \mathcal{T}_k \mathcal{W} + \beta_k ||\mathcal{L}_k \mathcal{V} - \mathcal{L}_k \mathcal{W}|| \\
&\leq \mathcal{T}_k \mathcal{W} + \beta_k \gamma_k ||\mathcal{V} - \mathcal{W}||
\end{aligned}
\tag{11}
$$

The first inequality follows from the properties of the sup-norm, the second inequality follows from the expectancy operator having a Lipschitz constant of one, and the final inequality follows from the properties of the Lipschitz operator $\mathcal{L}_k$. From Eq (11), we have

$$
\mathcal{T}_k \mathcal{V} - \mathcal{T}_k \mathcal{W} \leq \beta_k \gamma_k ||\mathcal{V} - \mathcal{W}||
$$

We can revert the roles of $\mathcal{V}$ and $\mathcal{W}$ in Eq (11) to obtain

$$
\mathcal{T}_k \mathcal{W} - \mathcal{T}_k \mathcal{V} \leq \beta_k \gamma_k ||\mathcal{V} - \mathcal{W}||
$$

We can thus conclude that

$$
|\mathcal{T}_k \mathcal{V} - \mathcal{T}_k \mathcal{W}| \leq \beta_k \gamma_k ||\mathcal{V} - \mathcal{W}||
\tag{12}
$$

Inequality (12) holds for all values of $k$, and we have

$$
||\mathcal{T}\mathcal{V} - \mathcal{T}\mathcal{W}|| \leq \eta ||\mathcal{V} - \mathcal{W}||
\tag{13}
$$

where $\eta \triangleq \max\{\beta_k \gamma_k | k = 1, \ldots, N\}$. That is, $\mathcal{T}$ is a contraction operator if $\eta < 1$. Q.E.D.

If $\mathcal{T}$ is operating on continuous functions on a compact state space, then it has a unique fix-point. Because $\mathcal{T}$ is a contraction, the fix-point can be obtained by iterations.

For our result to apply to a periodic problem, it remains to be shown that the equation set (5) is a special case of (6) and that the requirement on $\eta$ holds. By definition, the left-hand sides of (5) and (6) are identical. We thus need to show that the right-hand side in (5), for all values of $k$, is a special case of (7), which defines the right-hand side of (6). Because we have proved that the stochastic formulations in (8) and (9) are also contractions, our result also applies to the stochastic analogous extensions of equation set (5). We summarize this result in the following corollary:

**Corollary:** The periodic optimization problem in equation set (5) and analog stochastic problems are contraction problems having unique solutions, that is, the value functions.

*Proof*: The operator defined by $\mathcal{L}_k V \triangleq V_i$ for all values of $k$, with $i = k+1$ for $k \in (1, \ldots, N-1)$ and $i = 1$ for $k = N$, is a Lipschitz operator with Lipschitz constant $\gamma_k = 1$. That is, (5) is a special case of (6). The parameter $\beta_k$ in (5) is a discount factor, and for all values of $k$ we have $\beta_k < 1$. Thus, $\eta < 1$, and the corollary follows from the theorem. Q.E.D.

The proof of the corollary can be readily modified to show that the classical Bellman problem (that is, set $i = k$ for all $k$ in the proof) is also a special case of (6), as is any choice for $i \in \{1, \ldots, N\}$. Furthermore, there exists a large set of Lipschitz continuous operators that fulfill the requirements of the theorem, and there are many potential applications of (6).

Note the use of the sup-norm in the theorem above. It represents a type of worst-case scenario regarding convergence (that is, no single point in state space can "hang out"). Thus, in many applications, we expect convergence to be faster than that implied by $\eta$.

A varying interval length requires suitable adaptions of $\Pi_k$, $F_k$, $X$, and $Y_k$ (or the comparable stochastic elements). If discounting is uniform in time and interval $k$ represents a share $\delta_k$ of

the $N$-cycle, such that $t_k - t_{k-1} = \delta_k \cdot (t_N - t_0)$, we have $\beta_k = \beta^{\delta_k}$. In many applied settings, the $N$-cycle represents a year, and $\beta$, the discount factor over $N$ intervals, is then the annual discount factor. The extension to varying interval length is important, not least because it allows for reductions in dimensionality. For example, consider a problem that is formulated on an annual level, but where one month is different such that the problem is periodic. Without the option of varying the interval length, the model will require $N = 12$ to rigorously capture the periodic feature. With varying interval length, $N = 2$ suffices.

## 3. Examples of applications

In this section, we provide two examples that illustrate the use of our method and the proposed numerical scheme. We first return to the problem that led to the above developments: a dynamic, infinite-horizon, discounted discrete-time optimization problem with a periodic feature. In addition to demonstrating our method, the first example also shows the relevance and potential importance of considering periodic features in operational decision problems. The example has two intervals with different prices. In the second example, we consider a simple, dynamic game where two agents with different production parameters supply a good to a common market. This example can be solved exactly and we illustrate that the iterative solution converges to the exact solution. It also illustrates a different type of application than that in the first example.

### 3.1 Periodic price in a resource model

We consider a management model for a renewable capital stock with periodic variations in the price parameter. The question becomes how to shift resource extraction toward the high-price intervals to maximize the net present value of returns. We compare the optimal periodic solution with solutions for models where the price is assumed constant and nonperiodic. The equation governing the stock evolution is

$$x_{k+1} = F(x_k) - u_k \tag{14}$$

where $x_k$ is the capital level at the beginning of interval $k$, and $u_k$ is the level of exploitation. Natural growth, represented by the function $F(x_k) = \delta x_k / (1 - x_k(1-\delta)/\omega)$, with $\delta = 4$ and $\omega = 1$, is a version of the Beverton–Holt growth function. $\delta$ is the growth rate and $\omega$ is the saturation point where $F(x_k) = x_k$. Beyond the saturation point ($x_k > \omega$), the natural surplus growth ($F(x_k) - x_k$) is negative. The growth function is identical in all intervals and thus carries no interval subscript. The return function for interval $k$ is a constant relative risk aversion utility function:

$$\Pi_k(x_k, u_k) = p_k \frac{u_k^{1-\gamma}}{1 - \gamma} \tag{15}$$

The parameter $p_k$ is an interval-specific price, and $\gamma$ measures the degree of relative risk aversion. The example has two intervals with different prices, where $p_1 = 1.0$ and $p_2 = 0.2$. Furthermore, $\gamma = 1/2$. We consider the capital left for future growth, $y_k = x_{k+1}$, as our decision variable. That is, $u_k$ is eliminated with $u_k = F(x_k) - y_k$. With the price entering linearly as in Eq (15), the decision is independent of price in the nonperiodic problem (that is, if $p_1 = p_2$, the price parameter can be factored out of the decision problem). Thus, the periodic problem poses a more complex problem than that posed by the "associated nonperiodic problems."

We solve equation set (5) numerically, subject to Eqs (14) and (15), and derive periodic optimal feedback decision rules as functions of the capital level at the beginning of each interval: $y_1(x)$ and $y_2(x)$. Fig 2 reports these decision rules together with the replacement curve (the 45-degree line, $y = x$). If the curves of the decision rules are below the replacement curve, the

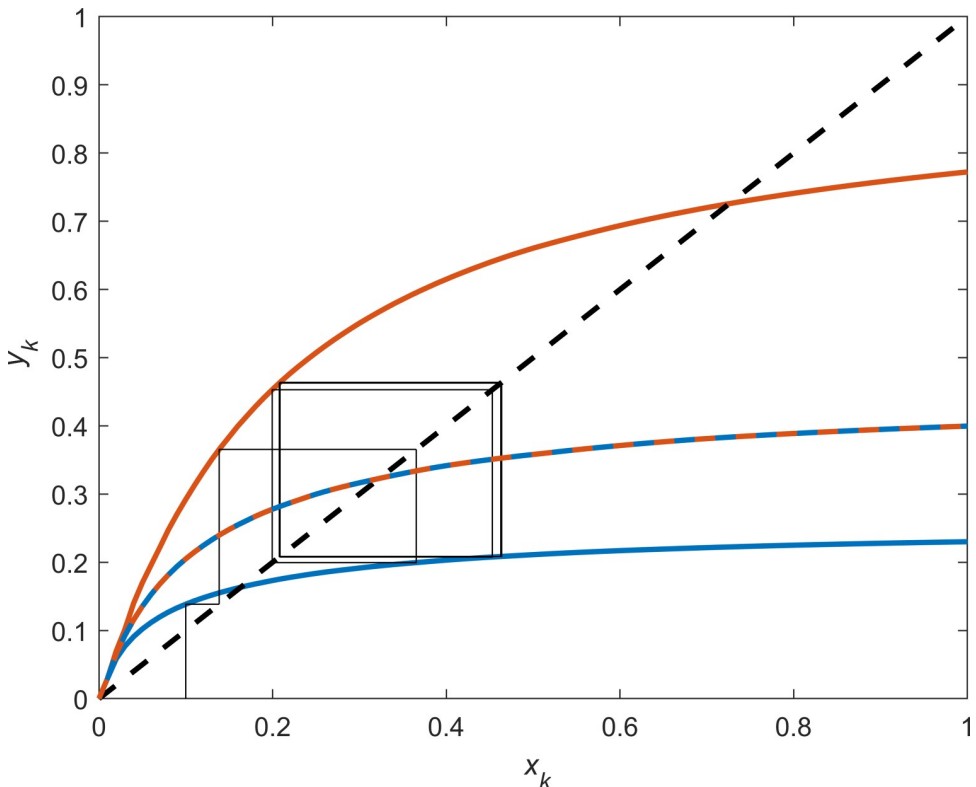

**Fig 2. Decision rules for the periodic problem example.** Decision rules for the periodic problem (blue solid curve $y_1$, red solid curve $y_2$), nonperiodic problems with prices 1.0 (blue dashed curve) and 0.2 (red dashed curve), replacement curve (black dashed curve), and sample path with initial $x_0 = 0.1$ (thin black path).

capital level is effectively reduced in the given interval and for the given capital level. The replacement curve also serves as the identity map used to transfer the state between subsequent periods ($y_k = x_{k+1}$). For comparison, the figure also reports the solutions to the two nonperiodic problems (with $p_1 = p_2$, one case with prices equal to 1.0 and the other with prices equal to 0.2). Because the price enters linearly in the objective function, the solutions to the nonperiodic problems are, as mentioned above, independent of the price and thus identical. Finally, the figure reports a sample time path for the stock level under the periodic solution, tracing out the stock level from interval to interval.

The decision rules for the periodic problem show significant and considerable discrepancies between intervals. In intervals with a high price ($p_1$), relatively small levels of capital are left for future growth (see the blue curve in Fig 2). Contrarily, in intervals with a low price ($p_2$), relatively high levels of capital are left for the future (see the red curve in Fig 2). Again, when the price is nonperiodic, the decision rule is invariant to the price level and, here, lies between the periodic decision rules (see the dashed red and blue curves in Fig 2). The sample time path starts at a chosen initial stock level ($x_0 = 0.1$), moves to the stock level in the first interval on the identity map via the first interval decision rule ($x_1 = y_1(x_0) = 0.138$), then moves to the stock level in the second interval via $y_2$ ($x_2 = y_2(x_1) = 0.365$), then moves to the stock level in the third interval via $y_1$ again ($x_3 = y_1(x_2) = 0.199$), and so on. The sample time path shows that the system under the periodic solution moves toward a stable two-period limit cycle and exhibits a more complex behavior than what can be discerned from the two nonperiodic problems that are associated with the periodic problem. The stock values of the limit cycle are

approximately $x_k = 0.208$ at interval 1 times (high price) and $x_k = 0.463$ at interval 2 times (low price). In the associated nonperiodic problems with both the high and the low price, the stock level approaches $x_k = 0.323$.

This example suggests that taking periodicity into account has significant practical implications. A more elaborate case was studied by Ni and Sandal [4], who examined a commercial fishery management problem involving a seasonal and regional separation of the spawning stock from the remaining stock. The problem is a two-state case with two intervals of 3 and 9 months. They implemented a special case of the present approach and were the first ones to demonstrate that a no-harvest region and a seasonal closure can develop naturally as a consequence of a first-best feedback policy. Another example was examined in Kvamsdal *et al.* [1], which demonstrated the potential pitfalls of a heuristic approximation of a periodic feature in a dynamic decision problem.

## 3.2 A dynamic game of supply

In our second example, we illustrate a different type of application than that considered in the first example. We consider a simple dynamic game where two agents with different cost parameters supply a good to a common market. In this market, the agents face a price that they both influence through their supply. The agents thus play a game where the optimal supply of each agent depends on the supply of the opposite agent. The example is designed such that it can be solved exactly for a given set of parameter values. We use this feature to compare the iterative solution—based on our theorem—with the exact solution.

The index $k$ has a different role in this example than in the periodic problem formulation (Fig 1). In the periodic problem, $k$ refers to time intervals; however, in this dynamic game problem, $k$ refers to agents. As we will see, the equations governing the solution of the dynamic game have the same structure as that of the equations governing the solutions of periodic problems, and thus, our approach is applicable.

Each agent $k$ ($k = 1,2$) owns a capital stock $x_k$, which has the following dynamics:

$$x_{k,t+1} = a + bx_{k,t} - u_k \tag{16}$$

The production $u_k$ is supplied to the market where it obtains the price $1 - \sum_i u_i$, $i = 1,2$. That is, the common price depends on the total supply to the market. The agents have private costs $C_k(x_k) = c_k + d_k x_k$. Then, the objective function of agent $k$ is revenues minus costs, $\Pi_k(x_k, u) = (1 - \sum_i u_i) u_k - C_k(x_k)$. Agent $k$ solves the following problem:

$$V_k(x) = \max_{u_k \le a+bx} \{\Pi_k(x, u) + \beta_k V_k(x')\}, \quad x' = a + bx - u_k, \quad k \in \{1, 2\} \tag{17}$$

That is, each agent maximizes the net present value of present returns, $\Pi_k(x, u)$, where the price they obtain depends on the total market supply, plus the discounted future value of their capital stock. Eq (17) defines an equation set that can be solved using our iterative approach.

The significance of this example is that we can obtain a formal solution that yields an exact solution for certain parameter values. Thus, we can confirm that our contraction approach yields the correct solution. To obtain the formal solution, we make an educated guess that the value functions take the quadratic form

$$V_k(x) = \theta_{1,k} x_1^2 + \theta_{2,k} x_1 x_2 + \theta_{3,k} x_2^2 + \theta_{4,k} x_1 + \theta_{5,k} x_2 + \theta_{6,k} \tag{18}$$

where $\theta_{i,k}$ are coefficients to be determined. Value functions of the second order is reasonable because the involved expressions (objective functions and dynamic constraints) are at most of the second order.

The first-order conditions can be solved to derive expressions for the decision rules of the agents, yielding a solvable set of equations for the coefficients $\theta_{i,k}$. That is, we take the derivative of the argument of the maximum operator in (17), substitute in Eqs (16) and (18), and put the resulting expressions equal to zero. These equations provide a solvable set of equations for the coefficients in Eq (18). The expression for $V_k(x)$ in (18) has six unknown coefficients for each of the two values of $k$, and we can derive 12 equations that define the solution. The first-order conditions also provide expressions for the decision rules $u_k$ defined in terms of the coefficients $\theta_{i,k}$. The resulting algebra is of limited interest; the interested reader can, in the Supplemental material, find pseudocode to solve the problem in Maple (or the algebraic solver of choice). Below, we only report the exact solution and convergence measures for the iterative solution.

For a numeric illustration, we choose the following set of parameters:

$$[a, b, c_1, c_2, d_1, d_2, \beta_1, \beta_2] = [0.5, 0.5, 0.4, 0.3, 0.2, 0.1, 0.95, 0.8]$$

For these parameters, we obtain the following exact solution (reporting only the first two non-zero digits):

$$V_1(x) = 0.019x_1^2 - 0.0095x_1x_2 + 0.0012x_2^2 + 0.072x_1 - 0.018x_2 + 1.0$$
$$V_2(x) = 0.0041x_1^2 - 0.0090x_1x_2 + 0.0050x_2^2 - 0.046x_1 + 0.056x_2 + 0.39$$

(19)

The coefficients in Eq (19) show that the value function for each agent increases with their private capital stock and decreases with the capital stock of the opposite agent. The corresponding optimal decision variables are defined in terms of the coefficients in equation set (19) and are given as the following feedback formulas:

$$u_1(x) = 0.12x_1 - 0.030x_2 + 0.13$$
$$u_2(x) = -0.061x_1 + 0.064x_2 + 0.26$$

(20)

That is, the optimal supply of each agent increases with their own capital stock and decreases with that of the other agent. The two supply functions in (20) are illustrated in Fig 3. The numeric approximation obtained from the contraction scheme closely matches the solutions in equation sets (19) and (20). To obtain the contraction solution, we use a rough uniform grid ($50 \times 50$ on the unit square, $[0,1]^2$) and apply five main policy iterations, each followed by 1,000 value iterations. The largest numeric deviations from the exact solutions are as follows:

$$V_1 : 0.0029, \quad V_2 : 0.00029, \quad u_1 : 0.00032, \quad u_2 : 0.00038$$

These deviations are small with regard to the grid size, and we conclude that the iterative solution has converged to the exact solution.

## 4. Conclusions

We arrived at the above theorem while working on periodic optimization problems. The major innovation that facilitated our insights was the consideration of a vector function rather than a scalar value function. The use of a vector function and our theorem above may be useful in applications other than periodic optimization problems. In what follows, we will discuss some potential applications and how problems can be formulated for our method to apply. We presume here that (8) is a suitable definition of the contraction operator, but depending on the application, the definitions in (7) or (9) may be better suited.

An application closely related to periodic optimization problems is finite-time optimization problems. These problems are typically solved by backward induction, but such solutions may

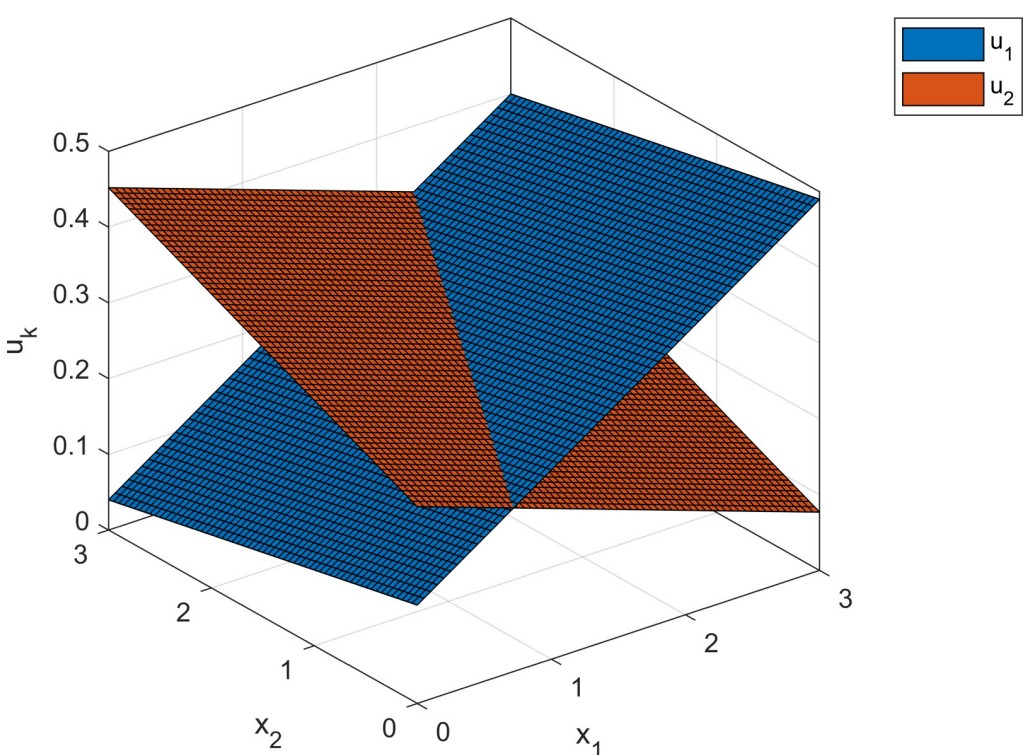

**Fig 3. Optimal supply for the dynamic game example.** Optimal supply for agent 1 (blue surface) and agent 2 (red surface) in the dynamic game.

be cumbersome to conciliate with given initial values. In contrast, our approach can be directly applied, where each interval is represented by an element in the vector function. Any form of nonautonomy can be accommodated (as with backward induction). Thus, for interval $k$, we have

$$V_k(x) = \max_{u \in \Gamma_k(x)} E_z\{\widehat{\Pi}_k(x, u, z) + \beta_k V_{k+1}(x')\}, \quad k \in \{1, \ldots, N\} \tag{21}$$

With $V_{N+1}(x) = G(x)$ representing the salvage value, the system of equations can be interpreted as a finite-time optimization problem with $N$ time intervals. Our corollary applies and shows that we have a contraction problem that, in general, can be solved. Solutions are on a general feedback form that is readily conciliated with a given initial value.

As one of our examples above demonstrates, some game theory problems can also be addressed using our methodology. We consider dynamic games over infinite time with non-cooperative (self-serving) behavior of $N$ agents, but where the decisions of other agents influence the return of each individual agent. Many common-pool resource games [15] fall within this type of games. For agent $k$, the problem is to maximize over one's own decisions while taking account of the decisions of others on both the current and future returns. Furthermore, the decisions may depend on, or be restricted by, a state vector $x$. Elements in the state vector may be common or private goods. The problem can be formulated as follows:

$$V_k(x) = \max_{u_k \in \Gamma_k(x)} E_z\{\widehat{\Pi}_k(x, u_k, u_{-k}, z) + \beta_k V_k(x')\}, \quad k \in \{1, \ldots, N\} \tag{22}$$

The notation $\widehat{\Pi}_k(x, u_k, u_{-k}, z)$ explicitly indicates that the return for agent $k$ depends on the

agent's own decisions ($u_k$) and those of all other agents ($u_{-k}$). The fact that the return function depends on the entire vector of decision variables ([$u_1$,. . .,$u_N$]) necessitates the consideration of a vector function ([$V_1$,. . .,$V_N$]). The proof of the corollary can be modified (with $i = k$) to show that this definition of the vector function can have a unique feedback solution. It relies on the specificities of the game and whether they imply the properties required for the various sets involved.

While known methods are applicable to decisions under uncertainty over future states, some decisions under risk of regime changes can be addressed using our approach. We think of regime changes as imposing significant changes to conditions for growth or production (utility). Say there are $N$ different regimes, and under a given regime, the return is given by $\widehat{\Pi}_k(x, u, z)$, while time evolution of the state variable may be regime-dependent and given by $F_k(x,u,z)$. Furthermore, let $\omega_k(x,u,z)$ denote a vector of probabilities for transitioning from regime $k$ to one of the $N$ regimes in the next period. These probabilities may differ under the different regimes and may further depend on the state variable, decision variable, or the stochastic component. Under regime $k$, the decision problem is as follows:

$$V_k(x) = \max_{u \in \Gamma_k(x)} E_z \left\{ \widehat{\Pi}_k(x, u, z) + \beta_k \sum_i \omega_{ki}(x, u, z) V_i(x') \right\}, \quad k \in \{1, \ldots, N\} \qquad (23)$$

Here, we sum over $i = 1..N$. The probabilities sum to one, such that $\beta_k < 1$ ensures that the above theorem holds. From this formulation, we see that the value for any given regime depends on a weighted sum of the elements in the value vector function. While the other suggested applications are clearly reminiscent of the original Bellman problem and can perhaps be perceived as "Bellman in a higher dimension," considering a linear combination of the value vector function is a fundamentally different structure.

A general solution method to solve periodic optimization problems is potentially a valuable tool in a wide range of settings. The renewable capital example above suggests that complex and atypical dynamics arise for a relatively modest deviation from the autonomous (nonperiodic) formulation. Fig 2 shows that the optimal periodic solution approaches a long-run limit cycle. Moreover, abstracting from periodicity—for example, by using heuristic approaches, such as considering an average effect rather than a periodic effect—can lead astray. Further examples show that such heuristics have severe, adverse consequences if management decisions are based on an autonomous approximation while agents, subject to these decisions, observe and adapt to the periodic phenomenon [1]. Inter-annual or within-season inefficiencies that agree well with these examples are observed in empirical studies of fisheries and have gained considerable attention [11, 12].

Periodicity in applied work in bioeconomics is often treated as short-term (intra-seasonal) dynamics, where considerable progress has been made in developing models to analyze inter-annual or within-season inefficiencies [16, 17]. In these models, natural population growth and discounting processes are frequently ignored. As indicated by Birkenbach et al. [17], these processes are more significant for the inter-season perspective, as developed in Kvamsdal et al. [1, 3]. Our results make it possible to merge these developments and develop models that account for intra- as well as inter-seasonal dynamics.

Our theorem is an intuitive extension of the Bellman result. The classical Bellman result is valid for a scalar value function. The periodic problems given in equation set (5) and implied by (7), (8), and (9) are non-autonomous, their value functions are autonomous vector functions, and the Bellman result is not applicable to these problems. But when the periodic cycle is perceived as the time unit, periodic problems can be perceived as autonomous in a higher dimension. As our suggestions for applications to dynamic games (equation set (22)) and

regime shifts (equation set (23)) demonstrate, our result applies to further problems with a genuinely different structure than that of the classical Bellman problem.

## Supporting information

**S1 File. Pseudo-code for Example 1.**
(PDF)

**S2 File. Code for Example 2.**
(PDF)

**S1 Appendix.**
(DOCX)

## Author Contributions

**Conceptualization:** Leif K. Sandal, Sturla F. Kvamsdal, José M. Maroto, Manuel Morán.

**Formal analysis:** Leif K. Sandal, Sturla F. Kvamsdal, José M. Maroto, Manuel Morán.

**Writing – original draft:** Leif K. Sandal, Sturla F. Kvamsdal, José M. Maroto, Manuel Morán.

**Writing – review & editing:** Leif K. Sandal, Sturla F. Kvamsdal, José M. Maroto, Manuel Morán.

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
