## [Decision Letter · Decision Letter 0]

16 Jun 2021

PONE-D-21-01335

A contraction approach to dynamic optimization problems

PLOS ONE

Dear Dr. Kvamsdal,

Thank you for submitting your manuscript to PLOS ONE. After careful consideration, we feel that it has merit but does not fully meet PLOS ONE’s publication criteria as it currently stands. Therefore, we invite you to submit a revised version of the manuscript that addresses the points raised during the review process.

We look forward to receiving your revised manuscript.

Kind regards,

Mohd Nadhir Ab Wahab, Ph.D.

Academic Editor

PLOS ONE

Journal Requirements:

2.In your Data Availability statement, you have not specified where the minimal data set underlying the results described in your manuscript can be found. PLOS defines a study's minimal data set as the underlying data used to reach the conclusions drawn in the manuscript and any additional data required to replicate the reported study findings in their entirety. All PLOS journals require that the minimal data set be made fully available. For more information about our data policy, please see http://journals.plos.org/plosone/s/data-availability.

3.Thank you for submitting the above manuscript to PLOS ONE. During our internal evaluation of the manuscript, we found significant text overlap between your submission and the following working paper, of you are also authors:

https://openaccess.nhh.no/nhh-xmlui/bitstream/handle/11250/2573862/A14_17.pdf?sequence=1&isAllowed=y

We note that this working paper appears to be copyrighted by KOPINOR. Please note that should your paper be accepted, all content including images, results, and duplicated text will be published under the Creative Commons Attribution (CC BY) 4.0 license, which means that they will be freely available online, and any third party is permitted to access, download, copy, distribute, and use these materials in any way, even commercially, with proper attribution. In order to publish any previously copyrighted material, PLOS ONE requires permission from the original copyright holder of the content to publish it under the CC BY 4.0 license.

Please clarify whether the authors have received written permission from KOPINOR to publish this content specifically under the CC BY 4.0 license and upload the granted permission to the manuscript as a supporting information file.

To seek permission from the copyright owner to publish this content under the Creative Commons Attribution License (CCAL), CC BY 4.0, please contact them with the following text and PLOS ONE Request for Permission form (http://journals.plos.org/plosone/s/file?id=7c09/content-permission-form.pdf):

“I request permission for the open-access journal PLOS ONE to publish XXX under the Creative Commons Attribution License (CCAL) CC BY 4.0 (http://creativecommons.org/licenses/by/4.0/). Please be aware that this license allows unrestricted use and distribution, even commercially, by third parties. Please reply and provide explicit written permission to publish XXX under a CC BY license.”

Please upload the granted permission to the manuscript as a Supporting Information file. In the figure caption of the copyrighted figure, please include the following text: “Republished from [ref] under a CC BY license, with permission from [name of publisher], original copyright [original copyright year].”

Please note that RightsLink permission forms often impose use restrictions that are incompatible with our CC BY 4.0 license, and we are therefore unable to accept these permissions. For this reason, we strongly recommend contacting copyright holders with the PLOS ONE Request for Permission form.

Additional Editor Comments:

Please address all the comments given by the reviewers.

Reviewers' comments:

Reviewer's Responses to Questions

**Comments to the Author**

1. Is the manuscript technically sound, and do the data support the conclusions?

Reviewer #1: Yes

Reviewer #2: Yes

2. Has the statistical analysis been performed appropriately and rigorously? 

Reviewer #1: N/A

Reviewer #2: Yes

3. Have the authors made all data underlying the findings in their manuscript fully available?

Reviewer #1: Yes

Reviewer #2: Yes

4. Is the manuscript presented in an intelligible fashion and written in standard English?

Reviewer #1: No

Reviewer #2: Yes

5. Review Comments to the Author

Reviewer #1: Abstract-

Add the evaluation and results achieved

Introduction-

some references are needed when term periodicity is discussed.

Try to avoid using 'see' a lot to refer to existing works.

Line107: () = (). (This should also be numbered as equation)

Line108: "...of problems of type (1)." <-- Unclear what type(1) means. Is it the equation?

Line117" "..classical outset yielding (1) is then.." again please mention Equation (1) so if it's the equation

Line121: "...The length of the cycle of intervals is then =..." - Show as Equation and number it

Line126: Fig1- the variable k does not appear in the figure. Why?

Also I suggest to have a clearer section to explain the steps taken in this research. It is a bit unclear generally.

Line233: The Section name can be improved. Separate the sections for the two examples or create a subsection, with clearer name of the examples being used.

Clearer explanation on the problem being used as example is needed.

Line287: For second example, can be put in different section. Further explanation of the problem is needed, before going into the technicalities.

Equation 15 not explained well.

Line298: Equation not numbered.

Line306: How are these set of parameters derived? Any reference on the values used here?

Equation 16 and 17 that follow after the parameter, are not well explained.

Line311: The values shown in Equation 18 do not indicate that (18) is an equation.

Outcome from Second Example is not very clear. Where is the outcome/results actually?

Line314: "We discovered the above results"... avoid these kind of phrase. Please point exactly what is referred or at which section

Line325&337&353: Number the equation

Lines 375-377: "As our suggestions for applications to dynamic games and regime shifts demonstrate, our result applies further to problems with a genuinely different structure than the classical Bellman problem" <-- Discuss this further. Why this is seen to be important discovery and how this will make you move forward with future related research?

OVERALL: The writing flow needs to be majorly improved. The structure (with sections) needs to be improved as well. It seems like the focus is put more on showing the formulas but not so much on the discussion and the steps taken on how these experiments are done.

Line401: More References needs to be added ~ at least 4-5 more recent papers

Reviewer #2: The authors present an extension of the classical Bellmann optimization algorithm to periodic control problems. The work is presented in a clear and technically sound manner. As my only minor comment, I suggest that the authors give a few more details how periodicity in optimal control problems is handled by now (they write "... is often treated in some ad-hoc manner" in the introduction). Also one or two references to such "ad-hoc" treatments would be appropriate in the first paragraph of the introduction.

6. PLOS authors have the option to publish the peer review history of their article (what does this mean?). If published, this will include your full peer review and any attached files.

Reviewer #1: No

Reviewer #2: No

---

## [Author Response · Author response to Decision Letter 0]

20 Jul 2021

Response to Reviewers

Manuscript PONE-D-21-01335

A contraction approach to dynamic optimization problems

First, let us express our gratitude towards the reviewers who have devoted time and effort to review our work. Their comments and suggestions have helped us improve our manuscript considerably. Below, we account for our revisions in context of the reviewer comments; our replies are marked as such. Comments of a technical nature, such as missing equation numbers and so on, are taken care of but omitted from the list below.

Reviewer #1: Abstract-

Add the evaluation and results achieved

Reply: We have revised the abstract. Our main finding, that what we refer to as periodic problems are part of a more general class of solvable contraction problems, were already mentioned in the abstract. We have highlighted this further. In the revised abstract, we refer some of the results from the examples as well.

Introduction-

some references are needed when term periodicity is discussed.

Reply: We have added a reference to Kvamsdal et al. (2020), who also consider periodic problems, although in a narrower sense than what we do in the present manuscript. We would also like to point out that periodicity is considered a basic concept in dynamic systems. The third paragraph, beginning on line 54 in the revised manuscript, discusses the term and what we presently mean by it.

Try to avoid using 'see' a lot to refer to existing works.

Reply: We agree and have revised the manuscript accordingly.

Line108: "...of problems of type (1)." <-- Unclear what type(1) means. Is it the equation?

Reply: Yes, we mean the equation. We have revised the manuscript for clarity on this point.

Line117" "..classical outset yielding (1) is then.." again please mention Equation (1) so if it's the equation

Reply: Yes, again we mean the equation. We have revised the manuscript accordingly.

Line126: Fig1- the variable k does not appear in the figure. Why?

Reply: The variable k is the interval index. We agree that Figure 1 was unclear and have revised the figure to clarify the role of k.

Also I suggest to have a clearer section to explain the steps taken in this research. It is a bit unclear generally.

Reply: We agree that the manuscript would benefit from an overview of how it is organized. We have added, to the end of the introduction section, a paragraph that outlines the various steps in the manuscript. The paragraph starts on line 99 in the revised manuscript.

Line233: The Section name can be improved. Separate the sections for the two examples or create a subsection, with clearer name of the examples being used.

Reply: We have revised the section name and added subsections for each of the examples, in accordance with the suggestion from the reviewer.

Clearer explanation on the problem being used as example is needed.

Reply: We have added some further explanation of both examples. For the first example, we have also moved discussion of related work from the end of section 2 to the example subsection (3.1).

Line287: For second example, can be put in different section. Further explanation of the problem is needed, before going into the technicalities.

Reply: We agree that the second example required further explanation. The example is now more carefully explained before getting to the technicalities. We have also revised the example throughout.

Equation 15 not explained well.

Reply: Earlier equation (15), now equation (17), is now explained better. In particular, the potentially confusing factor, the different role of the index k from the previous example, is explained in a separate paragraph that starts on line 337 in the revised manuscript.

Line306: How are these set of parameters derived? Any reference on the values used here?

Reply: The set of parameters are simply chosen for numerical illustration. The manuscript has been revised to make this clear.

Equation 16 and 17 that follow after the parameter, are not well explained.

Reply: We have added brief explanations of the equations that now are numbered (19) and (20). Further, the supply functions in equation (20) are illustrated in the added Figure 3.

Line311: The values shown in Equation 18 do not indicate that (18) is an equation.

Reply: We agree that this is not an equation and have revised the manuscript accordingly.

Outcome from Second Example is not very clear. Where is the outcome/results actually?

Reply: In response to several of the comments from the reviewer, the second example is now better explained, and we hope the outcomes are clearer. We have illustrated the optimal decision rules in the new Figure 3. Let us hasten to add that the main purpose of the example was to show that the iterative solution converged to the exact solution. The example has been further revised on this point.

Line314: "We discovered the above results"... avoid these kind of phrase. Please point exactly what is referred or at which section

Reply: We had the main theorem in mind and have revised the manuscript accordingly.

Lines 375-377: "As our suggestions for applications to dynamic games and regime shifts demonstrate, our result applies further to problems with a genuinely different structure than the classical Bellman problem" <-- Discuss this further. Why this is seen to be important discovery and how this will make you move forward with future related research?

Reply: We found this suggestion a bit confusing. The paragraph was meant to briefly summarize the discussion of the various applications that we discuss earlier in the section. What we have done in our revision, however, is to add a brief paragraph prior to this final paragraph. In the added paragraph, we discuss work in bioeconomics that could be directly influenced by our contribution.

OVERALL: The writing flow needs to be majorly improved. The structure (with sections) needs to be improved as well. It seems like the focus is put more on showing the formulas but not so much on the discussion and the steps taken on how these experiments are done.

Reply: We have revised the manuscript to improve the writing flow. Several of the revisions resulting from other, more specific comments from both reviewers have also contributed to an improved flow. Finally, we have had the manuscript processed by a professional copyeditor.

Line401: More References needs to be added ~ at least 4-5 more recent papers

Reply: Four references from the last decade has been added at various places throughout the manuscript.

Reviewer #2: The authors present an extension of the classical Bellmann optimization algorithm to periodic control problems. The work is presented in a clear and technically sound manner. As my only minor comment, I suggest that the authors give a few more details how periodicity in optimal control problems is handled by now (they write "... is often treated in some ad-hoc manner" in the introduction). Also one or two references to such "ad-hoc" treatments would be appropriate in the first paragraph of the introduction.

Reply: While we never meant it that way, we realize that the term ‘ad-hoc’ could be understood as criticism. The point being that before a generic treatment like ours had been proposed, any approach would necessarily be ad-hoc in the literal sense of being for special purpose. We have thus avoided this term. Further, we have revised the first paragraph of the introduction and provide an example of practice that could be questioned. We also refer to some articles that illustrate, by mean of examples, that proper modeling of periodicity in optimal control problems matter. Notably, we have kept this discussion brief at this point in the manuscript. The examples are discussed somewhat further in the examples section. In addition, a brief paragraph prior to the final paragraph has been added where we discuss work in bioeconomics that could be directly influenced by our contribution.

---

## [Decision Letter · Decision Letter 1]

8 Nov 2021

A contraction approach to dynamic optimization problems

PONE-D-21-01335R1

Dear Dr. Kvamsdal,

We’re pleased to inform you that your manuscript has been judged scientifically suitable for publication and will be formally accepted for publication once it meets all outstanding technical requirements.

Kind regards,

Mohd Nadhir Ab Wahab, Ph.D.

Academic Editor

PLOS ONE

Additional Editor Comments (optional):

Please address the comments given by the reviewers.

Reviewers' comments:

Reviewer's Responses to Questions

**Comments to the Author**

1. If the authors have adequately addressed your comments raised in a previous round of review and you feel that this manuscript is now acceptable for publication, you may indicate that here to bypass the “Comments to the Author” section, enter your conflict of interest statement in the “Confidential to Editor” section, and submit your "Accept" recommendation.

Reviewer #1: All comments have been addressed

Reviewer #2: All comments have been addressed

2. Is the manuscript technically sound, and do the data support the conclusions?

Reviewer #1: Yes

Reviewer #2: (No Response)

3. Has the statistical analysis been performed appropriately and rigorously? 

Reviewer #1: N/A

Reviewer #2: (No Response)

4. Have the authors made all data underlying the findings in their manuscript fully available?

Reviewer #1: Yes

Reviewer #2: (No Response)

5. Is the manuscript presented in an intelligible fashion and written in standard English?

Reviewer #1: Yes

Reviewer #2: (No Response)

6. Review Comments to the Author

Reviewer #1: Revision has addressed the given comments. From the revised version, some improvements can be done:

Abstract - avoid starting first sentence with "We..." - try to rephrase some of the sentences in the abstract into a passive tone. Strongly recommend abstract to be rewritten.

Some of the section heading can be made clearer:

3. Examples (rename this section) - 3. Examples of....?

4.Final remarks -- suggest to be renamed as Conclusion

Reviewer #2: (No Response)

7. PLOS authors have the option to publish the peer review history of their article (what does this mean?). If published, this will include your full peer review and any attached files.

Reviewer #1: No

Reviewer #2: **Yes: **Volker Ahlers

---

## [Editor Report · Acceptance letter]

12 Nov 2021

PONE-D-21-01335R1 

A contraction approach to dynamic optimization problems 

Dear Dr. Kvamsdal:

I'm pleased to inform you that your manuscript has been deemed suitable for publication in PLOS ONE. Congratulations! Your manuscript is now with our production department. 

Kind regards, 

on behalf of

Dr. Mohd Nadhir Ab Wahab 

Academic Editor

PLOS ONE